# 3DPE-Gaze:Unlocking the Potential of 3D Facial Priors for Generalized Gaze Estimation

**Yangshi Ge     Yiwei Bao     Feng Lu**[*]
State Key Laboratory of VR Technology and Systems, School of CSE, Beihang University
{geyangshi, baoyiwei, lufeng}@buaa.edu.cn

## Abstract

In recent years, face-based deep-learning gaze estimation methods have achieved significant advancements. However, while face images provide supplementary information beneficial for gaze inference, the substantial extraneous information they contain also increases the risk of overfitting during model training and compromises generalization capability. To alleviate this problem, we propose the 3DPE-Gaze framework, explicitly modeling 3D facial priors for feature decoupling and generalized gaze estimation. The 3DPE-Gaze framework consists of two core modules: the 3D Geometric Prior Module (3DGP) incorporating the FLAME model to parameterize facial structures and gaze-irrelevant facial appearances while extracting gaze features; the Semantic Concept Alignment Module (SCAM) separates gaze-related and unrelated concepts through CLIP-guided contrastive learning. Finally, the 3DPE-Gaze framework combines 3D facial landmark as prior for generalized gaze estimation. Experimental results show that 3DPE-Gaze outperforms existing state-of-the-art methods on four major cross-domain tasks, with particularly outstanding performance in challenging scenarios such as lighting variations, extreme head poses, and glasses occlusion.

## 1   Introduction

Gaze estimation has wide applications in computer vision, being an essential technology in scenarios such as human-computer interaction[13, 28, 29], virtual/augmented reality[25, 31], and driving monitoring[1, 19, 27]. Gaze estimation methods can be broadly categorized into two types: model-based approaches and appearance-based approaches. Model-based approaches calculate gaze direction by simulating the anatomical structure of the eyeball, offering higher accuracy but typically requiring specialized hardware and controlled environments. Appearance-based approaches, on the other hand, learn gaze mapping relationships directly from image features, offering broader application potential.

Appearance-based gaze estimation research has undergone a critical transition from focusing solely on eye regions to utilizing full-face information. Early research primarily extracted local features from eye regions[26, 20], but this approach overlooked the important contextual information provided by other facial areas. Zhang et al.[38] pioneered the use of full-face input and introduced a spatial weighting mechanism, significantly improving prediction accuracy. Since then, full-face input has become the mainstream approach in gaze estimation research.

However, the full-face input strategy is a double-edged sword: while it contains rich gaze-related structural information, the eye region occupies only a small proportion of the image, simultaneously introducing numerous domain-specific interferences (such as lighting variations, skin color differences, expression changes, etc.)[33]. These irrelevant information easily lead to models overfitting to

---

[*]Corresponding author.

39th Conference on Neural Information Processing Systems (NeurIPS 2025).

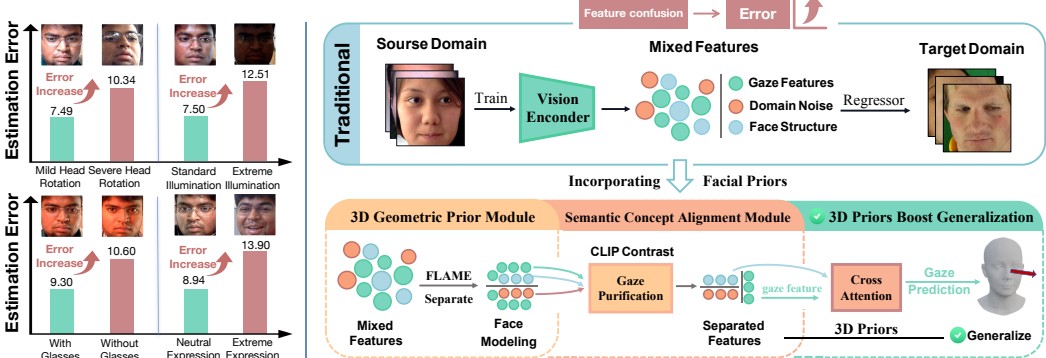

Figure 1: Left shows traditional CNN exhibiting clear performance collapse under standard conditions versus domain shift conditions. Right briefly demonstrates how we introduce 3D facial priors.

appearance features in the training data, severely affecting cross-domain generalization capability. To address this challenge, researchers have proposed various solutions, such as adversarial learning[7], data perturbation[33], feature separation[18], and multimodal fusion[36, 35]. However, most of these methods remain at the image feature level and fail to fully utilize 3D facial priors, making it difficult for them to capture the inherent geometric structure of the face and more susceptible to surface appearance changes. Therefore, **how to fully utilize structured facial information while effectively filtering domain-specific noise** remains a key challenge in the field of gaze estimation.

To address this challenge, we propose the Three-Dimensional Prior Enhanced Gaze Estimation framework (**3DPE-Gaze**), consisting of two complementary core modules: **3D Geometric Prior Module (3DGP)** and **Semantic Concept Alignment Module (SCAM)**. As shown in Figure 1, the 3DGP module decouples head pose, expression, shape, and gaze parameters from 2D facial images through the FLAME parametric face model, transforming gaze estimation from a prediction process based on domain-susceptible 2D appearance features to one based on stable geometric structures. However, FLAME-based encoders [9, 8]have inherent limitations in precisely modeling eyeball rotation and capturing subtle eye details, making it difficult to fully express complex gaze behaviors. To compensate for this deficiency, we designed the SCAM module as a complement. This module, based on CLIP semantic representations, distinguishes between gaze-relevant and irrelevant features through contrastive learning. Our FacePrior-Gaze Predictor effectively fuses these two complementary types of information, enabling the model to simultaneously leverage structured geometric constraints and high-level semantic concepts, thereby enhancing cross-domain generalization capability.

In summary, **3DPE-Gaze** systematically injects 3D facial priors and semantic concept priors into the gaze estimation process, effectively solving the cross-domain generalization problem in gaze prediction. Our experiments on multiple benchmark datasets have validated the excellent cross-domain performance of this method, achieving significant performance improvements in various challenging scenarios compared to current state-of-the-art methods. The main contributions of this research are as follows:

- We propose the 3D Geometric Prior Module (3DGP), a novel parametric encoder that transforms the gaze estimation problem from pixel space to parameters such as head pose, expression, and shape, establishing the foundation for associating gaze prediction with 3D facial structure and achieving effective extraction of domain-invariant features.

- We propose the Semantic Concept Alignment Module (SCAM), specifically addressing the limitations of pure geometric modeling in capturing complex gaze semantics. This module innovatively leverages CLIP pre-trained knowledge and contrastive learning strategies to explicitly distinguish gaze-relevant concepts from domain interference factors in the feature space, allowing the model to adapt to new environments without any target domain data.

- Experimental results show that our 3DPE-Gaze framework achieves significant improvements in various cross-domain settings. In cross-dataset evaluations, it reduces average error by more than 27% compared to the baseline and improves by 6% over SOTA methods, with particularly outstanding performance in highly challenging scenarios.

## 2 Related Work

**Gaze Estimation.**  Gaze estimation has shifted from using local eye features [26, 20] to full-face inputs [38], which, despite including richer structural information, introduced domain-specific noise that impairs generalization. Existing solutions have largely focused on either purifying 2D image features [7, 33, 18] or introducing prior knowledge. For instance, some methods incorporate geometric cues like head pose as auxiliary information [40, 38, 22, 32, 17]. However, these approaches often treat geometric priors as supplementary inputs rather than core representations. In contrast, our 3DGP module fundamentally reframes the problem by using a parametric 3D model to map faces from an unstructured pixel space to a physically meaningful and decoupled parameter space. This transforms the task from simple "pixel-to-gaze" regression into a more robust "structure-to-gaze" prediction.

Other works have leveraged large-scale pre-trained models like CLIP to enhance robustness [35, 36]. These methods typically apply visual-linguistic knowledge to general 2D image features. Our work differs significantly in its design. The SCAM module performs a targeted semantic purification on a specific gaze code that has already been geometrically isolated by the 3DGP module. This novel "secondary purification" on a pre-decoupled feature is a key contribution of our framework.

**Face Parametric Model.**  3D face models have evolved from the 3DMM[2] first proposed by Blanz and Vetter, to BFM[23], FaceWarehouse[3], and then to FLAME[16], gradually enhancing the modeling capabilities for facial shape, expression, and movement.  Although recent models like ICT[15] and FaceScape[34] offer higher geometric detail in appearance, they lack FLAME's capabilities in feature disentanglement and efficiency. FLAME's key advantage lies in its explicit parametric decoupling of shape, pose, and expression, a characteristic that highly aligns with the need in gaze estimation to separate gaze-related features from domain-specific interference. Additionally, it achieves a good balance between expressiveness and computational efficiency.

## 3 Method

### 3.1 3DPE-Gaze Framework Overview

To unlock the potential of 3D facial priors for generalized gaze estimation, we propose the **3DPE-Gaze** framework. As shown in Figure 2, the 3DPE-Gaze framework consists of two core modules and a final gaze predictor:

**3D Geometric Prior Module (3DGP)**   The 3DGP module, based on the parametric face model FLAME, decomposes facial images into shape, expression, pose, lighting, and gaze parameters, achieving explicit decoupling of gaze features, facial structure features, and domain interference factors in physical space. This module also provides physically constrained anatomical foundations for gaze estimation through the reconstruction of 3D facial structures.

**Semantic Concept Alignment Module (SCAM)**   The SCAM module utilizes CLIP pre-trained knowledge and contrastive learning strategies. This module explicitly distinguishes between "gaze-related concepts" (such as looking up-left) and "domain interference concepts" (such as lighting variations, facial expressions, etc.) in the feature space, providing cross-domain stable representations at the semantic level.

**Facial Prior-Gaze Predictor**   The Facial Prior-Gaze Predictor integrates facial structural features extracted by 3DGP with semantic concepts identified by SCAM. This decoder focuses on highly gaze-relevant geometric regions through attention mechanisms while suppressing domain-specific interference such as background and lighting, generating the final accurate gaze prediction results.

This design successfully unlocks the potential of 3D facial priors in gaze estimation: the 3DGP module provides stable facial prior constraints, while the SCAM module compensates for FLAME's limitations in eyeball modeling, offering semantic-level supplements. The complementary fusion of both significantly enhances the model's cross-domain generalization capability. Technical details of each module will be elaborated in the following sections.

### 3.2 3D Geometric Prior Module (3DGP)

The 3DGP module decomposes input images into structured geometric parameter representations through a parametric face model, thereby transferring gaze estimation from the pixel space, which

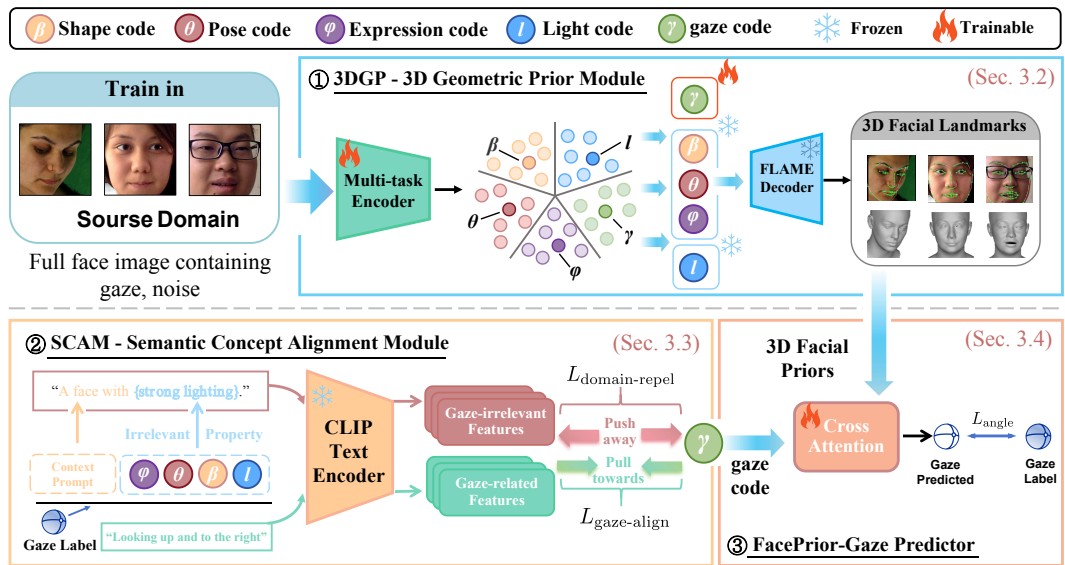

Figure 2: Overview of the proposed 3DPE-Gaze framework. The framework contains two core modules: (1) The 3D Geometric Prior Module (3DGP) utilizes the FLAME model to decompose facial images into geometric parameters, achieving physical decoupling of gaze features and domain interference; (2) The Semantic Concept Alignment Module (SCAM) separates gaze-related and unrelated concepts through CLIP-guided contrastive learning.

is susceptible to domain interference, to a more stable, decoupled parameter space. Through this decomposition, we can explicitly separate gaze code, facial structure parameters that affect gaze appearance, and domain-specific lighting parameters.

**Parametric Representation and Encoding**: Based on the FLAME model[16], we designed a gaze-specific multi-task encoder that receives preprocessed facial images II, and outputs a set of geometric parameters:

$$[\beta, \theta, \psi, \gamma, l] \tag{1}$$

where, $\beta \in \mathbb{R}^{100}$ controls static facial shape, $\theta \in \mathbb{R}^6$ represents head pose, $\psi \in \mathbb{R}^{50}$ captures dynamic expressions, $\gamma \in \mathbb{R}^{256}$ is a gaze-specific encoding containing eyeball movement information, and $l \in \mathbb{R}^{27}$ represents lighting parameters modeling environmental light conditions.

This parametric design decomposes facial images into domain-invariant geometric factors and susceptible appearance factors. For example, head pose $\theta$ and expression parameters $\psi$ remain relatively consistent under different lighting or skin color conditions, while lighting parameters $l$ explicitly capture environmental variation information. This explicit decoupling significantly reduces the model's dependence on domain-specific appearance features.

To provide stable and structurally constrained facial parametric representations, we reused the encoder architecture of the DECA[9] pre-trained model (based on a ResNet backbone) and initialized all layers corresponding to the $\beta, \theta, \psi, l$ parameters in the multi-task encoder with its pre-trained weights (including the shared ResNet backbone and corresponding output branches). During training, these DECA-initialized parts are completely frozen (non-trainable) to ensure the accuracy and stability of facial structure parameters. In contrast, only the encoding branch responsible for outputting the gaze code $\gamma$ is trainable. Considering that the $\gamma$ parameter is a specific mapping of generic features extracted by the ResNet backbone to gaze direction, we designed this branch as a multi-layer perceptron (MLP) connected after the shared ResNet backbone output features, able to directly learn the mapping from robust features to gaze parameters, focusing on gaze feature learning.

**3D Reconstruction and Structural Constraints**: After obtaining FLAME parameters, we reconstruct the 3D facial mesh through the FLAME decoder:

$$M_{\text{pred}} = \text{FLAME}(\beta, \theta, \psi) = \bar{T} + B_S(\beta) + B_P(\theta) + B_E(\psi, \theta) \tag{2}$$

where $\bar{T}$ is the average face template, $B_S(\beta)$ is the shape blend shape controlled by identity parameters, $B_P(\theta)$ is the pose-related deformation, and $B_E(\psi, \theta)$ is the expression blend shape. From the reconstructed facial mesh, we extract key 3D facial landmarks $K \in \mathbb{R}^{N \times 3}$, which provide anatomy-based constraints for subsequent gaze decoding, ensuring that gaze predictions conform to physical laws.

Compared to traditional methods, the 3DGP module transforms gaze estimation from a black-box "pixel-to-gaze" mapping to a more structured "structure-to-gaze" prediction. By decoupling known geometric factors, it enhances model interpretability and significantly improves cross-domain stability.

### 3.3 Semantic Concept Alignment Module (SCAM)

The SCAM module compensates for 3DGP's limitations in semantic understanding by explicitly distinguishing between "gaze-related" and "gaze-irrelevant" features at the concept level through the powerful vision-language alignment ability of the CLIP[24].

**Systematic Generation of Semantic Concepts**: To provide robust semantic supervision, we employ a systematic methodology for generating language descriptions for both gaze-related ($\mathcal{S}_{\text{gaze}}$) and domain interference ($\mathcal{S}_{\text{domain}}$) concepts.

For gaze-related concepts, a template-based approach is utilized, conditioned on the ground-truth gaze direction. We discretize gaze directions into canonical zones (e.g., "upper-right," "lower-left"), from which textual prompts such as, "A person looking towards the upper-right," are generated. This ensures the semantic representation is dynamically and accurately aligned with the precise directional gaze information for each sample.

For domain interference concepts, we leverage the physical parameters already extracted by our 3DGP module to programmatically generate a diverse set of gaze-irrelevant descriptions. This strategy creates a strong, coherent link between our geometric decoupling and semantic purification stages. Specifically, these prompts describe: **(1) Lighting conditions**, derived from the light parameters $l$ (e.g., "under strong light," "in dim light"); **(2) Facial expressions**, based on the expression parameters $\psi$ (e.g., "a neutral expression," "a person with a significant facial expression"); **(3) Facial shape**, informed by the shape parameters $\beta$; and **(4) Head pose**, based on the pose parameters $\theta$ (e.g., "head tilted to the left").

These systematically generated sets of descriptions are then passed through the CLIP text encoder to obtain their semantic representations:

$$t_{\text{gaze}} = \Psi_{\text{CLIP}}(s_{\text{gaze}}), \quad s_{\text{gaze}} \in \mathcal{S}_{\text{gaze}} \tag{3}$$

$$t_{\text{domain}} = \Psi_{\text{CLIP}}(s_{\text{domain}}), \quad s_{\text{domain}} \in \mathcal{S}_{\text{domain}} \tag{4}$$

where $\Psi_{\text{CLIP}}(\cdot)$ represents the CLIP encoder.

**Contrastive Learning and Concept Separation**: To align gaze code $\gamma$ with gaze concepts in semantic space while keeping them away from domain interference concepts, we designed a bidirectional contrastive loss:

**Gaze Concept Alignment Loss**:

$$\mathcal{L}_{\text{gaze-align}} = -\log \frac{\exp(\text{sim}(z_\gamma, t_{\text{gaze}})/\tau)}{\sum_j \exp(\text{sim}(z_\gamma, t_j)/\tau)} \tag{5}$$

where $z_\gamma$ is the projected gaze parameter, sim calculates cosine similarity, and $\tau$ is a temperature parameter.

**Domain Interference Repulsion Loss**:

$$\mathcal{L}_{\text{domain-repel}} = \frac{1}{|\mathcal{S}_{\text{domain}}|} \sum_{s \in \mathcal{S}_{\text{domain}}} \max(0, \text{sim}(z\gamma, \Psi_{\text{CLIP}}(s)) - m) \tag{6}$$

where $m$ is a boundary parameter, ensuring that gaze features maintain sufficient distance from domain interference concepts.

Through this contrastive mechanism, the SCAM module establishes an explicit boundary between gaze features and domain interference features in semantic space, enabling the model to focus on

high-level concepts related to gaze while filtering out domain-specific interferences. Complementary to geometric priors, semantic priors demonstrate unique advantages in handling abstract visual concepts and complex visual interferences.

## 3.4 Facial Prior-Gaze Predictor

To fully utilize both geometric and semantic priors, we designed a collaborative decoder that makes the two representations mutually enhancing through an attention mechanism, jointly guiding gaze prediction.

**Cross-modal Feature Fusion**: We adopt a cross-attention mechanism, using semantically guided gaze code $\gamma$ as the Query, and three-dimensional geometric landmarks $K$ as the Key and Value:

$$F_{\text{fused}} = \text{CrossAttention}(Q = W_q\gamma, ; K = W_k K, ; V = W_v K) \tag{7}$$

where $W_q, W_k, W_v$ are learnable parameter matrices. This design enables the model to selectively focus on key structural regions based on semantic understanding, increasing sensitivity to gaze-related features.

**Gaze Vector Prediction**: The fused features generate gaze predictions through the final MLP decoder:

$$g = \frac{f_{\text{reg}}(F_{\text{fused}})}{|f_{\text{reg}}(F_{\text{fused}})|_2} \tag{8}$$

This collaborative mechanism achieves complementary enhancement of geometric and semantic priors: the structural features provided by 3DGP ensure that gaze predictions conform to physical constraints, while the semantic concepts contributed by SCAM guide the model to focus on the most relevant regions and suppress interference. Working together, they effectively alleviate the problem of traditional methods over-relying on domain-specific appearance features while fully utilizing structured information within the full-face range.

## 3.5 Training Objectives and Implementation

We adopt a multi-objective joint optimization strategy to train the 3DPE-Gaze framework, with the overall loss function:

$$\mathcal{L} = \lambda_1\mathcal{L}_{\text{angle}} + \lambda_2\mathcal{L}_{\text{gaze-align}} + \lambda_3\mathcal{L}_{\text{domain-repel}} \tag{9}$$

**Gaze Angular Loss** $\mathcal{L}$angle measures the angular difference between predicted gaze and true gaze:

$$\mathcal{L}_{\text{angle}} = \arccos\left(\frac{\hat{g}^T g}{||\hat{g}||_2||g||_2}\right) \tag{10}$$

**Gaze Concept Alignment Loss** $\mathcal{L}_{\text{gaze-align}}$ encourages gaze parameters $\gamma$ to align with gaze-related concepts.

**Domain Interference Repulsion Loss** $\mathcal{L}_{\text{domain-repel}}$ ensures that gaze parameters $\gamma$ stay away from interference concepts.

Hyperparameters $\lambda_1$, $\lambda_2$, and $\lambda_3$ are used to balance the contributions of each loss term. Empirically, we set $\lambda_1 = \lambda_2 = \lambda_3 = 1.0$.

**Training Implementation Details** We conducted experiments on a single NVIDIA A100 GPU. Specifically, we adopted a staged training strategy: first freezing the FLAME parameter branches and pre-training the gaze parameter $\gamma$; then introducing contrastive learning of the SCAM module and jointly optimizing the entire model. We used the Adam optimizer with an initial learning rate of $10^{-4}$ and a batch size of 256.

# 4 Experiments

**Datasets and Evaluation Protocol.** We adopt experimental settings consistent with cutting-edge research in cross-domain gaze estimation [7, 33, 36, 35, 18], evaluating our method on four

cross-domain tasks. Specifically, we use ETH-XGaze[39] and Gaze360[14] as training datasets, and MPIIFaceGaze[37] and EyeDiap[12] as testing datasets. For concise representation, we denote these four cross-domain tasks as $\mathcal{D}_E$(ETH-XGaze)$\rightarrow \mathcal{D}_M$(MPIIFaceGaze), $\mathcal{D}_E \rightarrow \mathcal{D}_D$(EyeDiap), $\mathcal{D}_G$(Gaze360)$\rightarrow \mathcal{D}_M$, and $\mathcal{D}_G \rightarrow \mathcal{D}_D$. This standardized cross-domain setup ensures that our experimental results can be directly compared with related research.

**Data Preprocessing.** For $\mathcal{D}_E, \mathcal{D}_M$, and $\mathcal{D}_D$, we normalize facial images following the standard method in [37]; for $\mathcal{D}_G$, we only select frontal face images to match the distribution characteristics of other datasets, which is consistent with former researches [7, 18]. All images are resized to a uniform resolution of 224×224 and normalized to the [0,1] range, thereby eliminating the influence of differences in acquisition devices and resolutions across different datasets.

### 4.1 Performance Comparison with State-of-the-Art Methods

Table 1: Performance comparison on cross-domain gaze estimation tasks (unit: degrees)

| Method | $\mathcal{D}_E \rightarrow \mathcal{D}_M$ | $\mathcal{D}_E \rightarrow \mathcal{D}_D$ | $\mathcal{D}_G \rightarrow \mathcal{D}_M$ | $\mathcal{D}_G \rightarrow \mathcal{D}_D$ | **Avg** |
|---|---|---|---|---|---|
| CNN Baseline | 8.56 | 8.90 | 9.51 | 8.48 | 8.86 |
| PureGaze [7] | 7.08 | 7.44 | 9.28 | 9.32 | 8.28 |
| CDG [30] | 6.73 | 7.95 | 7.03 | 7.27 | 7.25 |
| Xu et al. [33] | 6.50 | 7.44 | 7.55 | 9.03 | 7.63 |
| Liang et al. [18] | **5.79** | 6.96 | 7.06 | 7.99 | 6.95 |
| CLIP-Gaze [36] | 6.41 | 7.51 | 6.89 | 7.06 | 6.96 |
| LG-Gaze [35] | 6.45 | 7.22 | 6.83 | 6.86 | 6.84 |
| Our 3DPE-Gaze | 6.66 | **6.13** | **6.71** | **6.23** | **6.43** |

Table1 shows the performance comparison between 3DPE-Gaze and existing state-of-the-art methods on four cross-domain gaze estimation tasks. The results demonstrate that our method surpasses SOTA methods in 3 out of 4 cross-domain settings. From the overall performance, our approach exhibits stronger general generalization capability, achieving the lowest average error across all four cross-domain tasks, fully validating the excellent performance of our proposed framework in addressing cross-domain challenges. This result also confirms that effective utilization of facial priors can significantly enhance the cross-domain generalization capability of gaze estimation models.

Table 2: In-domain gaze estimation performance comparison (unit: degrees).

| Method | **within $\mathcal{D}_M$** | **within $\mathcal{D}_D$** | **within $\mathcal{D}_G$** | **within $\mathcal{D}_E$** |
|---|---|---|---|---|
| Dilated-Net [4] | 4.42 | 6.19 | 13.73 | N/A |
| Gaze360 [14] | 4.06 | 5.36 | 11.04 | 4.46 |
| RT-Gene [11] | 4.66 | 6.02 | 12.26 | N/A |
| FullFace [38] | 4.93 | 6.53 | 14.99 | 7.38 |
| RCNN [21] | 4.10 | 5.31 | 11.23 | N/A |
| CA-Net [6] | 4.27 | 5.27 | 11.20 | N/A |
| GazeTR-Pure [5] | 4.74 | 5.72 | 13.58 | N/A |
| GazeTR-Hybird [5] | 4.00 | 5.17 | 10.62 | N/A |
| CNN Baseline | 4.74 | 7.49 | 13.23 | 5.69 |
| Our 3DPE-Gaze | 4.03 | 5.06 | 11.83 | 4.39 |

While our 3DPE-Gaze framework is primarily designed to enhance cross-domain generalization, it is also crucial to validate that this improvement does not compromise its performance within a single domain. Therefore, we conducted in-domain experiments, with the results presented in Table 2. The results show that our method achieves highly competitive, and in some cases state-of-the-art, performance. This demonstrates that our approach of leveraging 3D facial priors not only significantly boosts cross-domain robustness but also maintains excellent accuracy for in-domain gaze estimation.

## 4.2 Diagnostic Analysis of Geometric Prior Quality

To understand the performance discrepancy across different tasks, particularly the weaker result on the $\mathcal{D}_E \to \mathcal{D}_M$ task, we conducted a diagnostic analysis. We hypothesize that the model's final accuracy is strongly correlated with the quality of the geometric priors extracted by the 3DGP module.

To test this, we use the stability of the reconstructed 3D Inter-ocular Distance (IOD) as a proxy for prior quality. For any given subject, their physical IOD is fixed; therefore, a lower standard deviation of the IOD across multiple images indicates a more stable and higher-quality 3D prior. As shown in Table 3, we grouped subjects from each target dataset into "High-Quality" and "Low-Quality" prior groups based on this metric.

The results clearly show that for both target domains, the High-Quality Prior group achieves significantly lower gaze error than the Low-Quality Prior group. This analysis confirms that our model's performance is indeed dependent on the quality of the extracted 3D geometry, and the instability of priors from the $\mathcal{D}_M$ dataset is the primary reason for the higher error in that specific cross-domain task.

Table 3: Diagnostic analysis of geometric prior quality and its impact on gaze error. Avg. IOD Std Dev refers to the average standard deviation of the reconstructed 3D Inter-ocular Distance.

| Domain | Prior Quality Group | Avg. IOD Std Dev | Average Gaze Error | Number of Subjects |
|--------|--------------------|-----------------|-------------------|--------------------|
| $\mathcal{D}_M$ | High-Quality Prior | 4.1 | 5.50° | 5 |
| | Low-Quality Prior | 10.5 | 7.24° | 10 |
| | Overall | 8.4 | 6.66° | 15 |
| $\mathcal{D}_D$ | High-Quality Prior | 3.8 | 5.38° | 9 |
| | Low-Quality Prior | 9.2 | 7.10° | 7 |
| | Overall | 6.2 | 6.13° | 16 |

## 4.3 Ablation Studies

**Effectiveness of Core Modules**

Our proposed framework relies on three complementary loss functions to achieve high-performance cross-domain gaze estimation: Gaze Angular Loss ($\mathcal{L}_{\text{angle}}$), Gaze Concept Alignment Loss ($\mathcal{L}_{\text{gaze-align}}$), and Domain Interference Repulsion Loss ($\mathcal{L}_{\text{domain-repel}}$). To verify the contribution of each loss function and their synergistic effect, we designed ablation experiments as shown in Table 4.

Table 4: Ablation experiments for different loss function combinations (unit: degrees)

| Model Configuration | $\mathcal{D}_E \to \mathcal{D}_M$ | $\mathcal{D}_E \to \mathcal{D}_D$ | $\mathcal{D}_G \to \mathcal{D}_M$ | $\mathcal{D}_G \to \mathcal{D}_D$ |
|---------------------|-----------------------------------|-----------------------------------|-----------------------------------|-----------------------------------|
| CNN Baseline | 8.56 | 8.90 | 9.51 | 8.48 |
| + $\mathcal{L}_{\text{domain-repel}}$ | 7.32 | 7.19 | 7.63 | 7.44 |
| + $\mathcal{L}_{\text{gaze-align}}$ | 8.03 | 8.35 | 8.40 | 7.87 |
| + SCAM ($\mathcal{L}_{\text{gaze-align}}$ + $\mathcal{L}_{\text{domain-repel}}$) | 7.18 | 6.86 | 7.32 | 7.17 |
| 3DGP Only ($\mathcal{L}_{\text{angle}}$) | 7.45 | 7.19 | 7.60 | 6.80 |
| + $\mathcal{L}_{\text{domain-repel}}$ | 6.90 | 6.50 | 7.05 | 6.35 |
| + $\mathcal{L}_{\text{gaze-align}}$ | 7.41 | 6.38 | 7.60 | 6.72 |
| + SCAM ($\mathcal{L}_{\text{gaze-align}}$ + $\mathcal{L}_{\text{domain-repel}}$) | **6.66** | **6.13** | **6.71** | **6.23** |

The results show that using the FLAME prior-based $\mathcal{L}_{\text{angle}}$ alone can significantly reduce errors, validating the fundamental value of geometric constraints. Further introducing $\mathcal{L}_{\text{gaze-align}}$ enhances model performance by leveraging semantic information. Finally, integrating $\mathcal{L}_{\text{domain-repel}}$ effectively separates gaze-related and unrelated features, bringing comprehensive performance improvements, with the complete model achieving the best average error. This indicates that the synergistic effect of geometric constraints, semantic guidance, and feature decoupling is crucial for improving cross-domain robustness.

**Impact of Backbone Model Choices**

To analyze the impact of our backbone model choices, we conducted ablation studies as suggested by reviewers, with results shown in Table 5. For the semantic encoder, we found that while more powerful CLIP architectures can improve performance, our chosen ViT-B-16 provides a strong balance between accuracy and efficiency. Similarly, for the FLAME parameter regressor, other high-quality models like SPECTRE also achieve competitive results, demonstrating our framework's flexibility. We selected DECA as our primary regressor due to its wide availability and recognized strong performance.

Table 5: Ablation study on the impact of different backbone models (CLIP text encoders and FLAME parameter regressors). Unit: degrees.

| Backbone Component | $\mathcal{D}_E \to \mathcal{D}_M$ | $\mathcal{D}_E \to \mathcal{D}_D$ | $\mathcal{D}_G \to \mathcal{D}_M$ | $\mathcal{D}_G \to \mathcal{D}_D$ |
|---|---|---|---|---|
| *Semantic Encoder (CLIP)* | | | | |
| RN50 | 6.90 | 6.55 | 6.95 | 6.44 |
| ViT-B-16 (Ours) | 6.66 | 6.13 | 6.71 | 6.23 |
| ViT-L-14 | 6.55 | 6.08 | 6.60 | 6.17 |
| ViT-H-14 | 6.50 | 6.05 | 6.61 | 6.12 |
| *FLAME Parameter Regressor* | | | | |
| EMOCA [8] | 6.85 | 6.40 | 6.90 | 6.45 |
| DECA (Ours) [9] | 6.66 | 6.13 | 6.71 | 6.23 |
| SPECTRE [10] | 6.59 | 6.10 | 6.65 | 6.18 |

**Optimization of Geometric Feature Transfer Paths**

Table 6: Ablation experiments for model architecture configurations (unit: degrees).

| Model Configuration | $\mathcal{D}_E \to \mathcal{D}_M$ | $\mathcal{D}_E \to \mathcal{D}_D$ | $\mathcal{D}_G \to \mathcal{D}_M$ | $\mathcal{D}_G \to \mathcal{D}_D$ |
|---|---|---|---|---|
| Landmarks Only | **6.66** | **6.13** | **6.71** | **6.23** |
| Landmarks + Pose | 7.20 | 6.70 | 6.85 | 6.47 |
| Landmarks + Full Parameters | 7.32 | 6.53 | 6.85 | 6.94 |

To identify the optimal geometric representation for generalization, we compared three feature configurations from the FLAME model: using only 3D landmarks, landmarks with head pose parameters, and landmarks with the full parameter set. As shown in Table 6, using only facial landmarks performed best across all tasks. This result suggests that landmarks provide a sufficiently structured abstraction of facial geometry, capturing key gaze-related relationships while filtering the domain-specific noise present in the more detailed parameters. Including the full parameter set introduced additional domain-specific biases that hindered performance, confirming that focusing on abstract, domain-invariant features is more effective for cross-domain learning.

## 4.4 Robustness Verification of Facial Priors in Extreme Scenarios

The core idea of our 3DPE-Gaze framework is to decouple gaze from other irrelevant features and incorporating 3D facial priors for generalized gaze estimation. Thus, in this section, we conduct experiments to verify the robustness of 3DPE-Gaze framework regarding varies factors, including head pose, lighting conditions, expression changes and glasses on the $\mathcal{D}_E \to \mathcal{D}_M$ task.

**Robustness under Extreme Lighting Conditions.** As shown in Figure 3 (left), our method outperforms the baseline under all lighting conditions, especially in extremes. It reduces errors by 8.3% in low-light and 10.6% in high-light areas, as our 3D geometric representation effectively separates facial structure from environmental lighting effects.

**Adaptability to Expression Changes.** Figure 3 (right) shows that our 3DPE-Gaze method consistently outperforms the baseline across the entire range of facial expressions. For extreme expressions (L1 intensity >17), our method maintains a stable error level while the baseline's error significantly increases. This is due to our model's ability to decouple expression muscle activity from gaze direction.

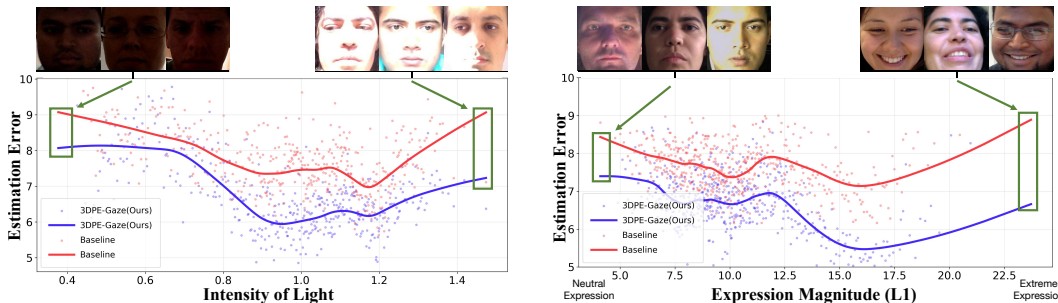

Figure 3: Robustness analysis under extreme scenarios. Left: Analysis of gaze estimation accuracy under different lighting intensities; Right: Impact of facial expression variations on gaze estimation precision.

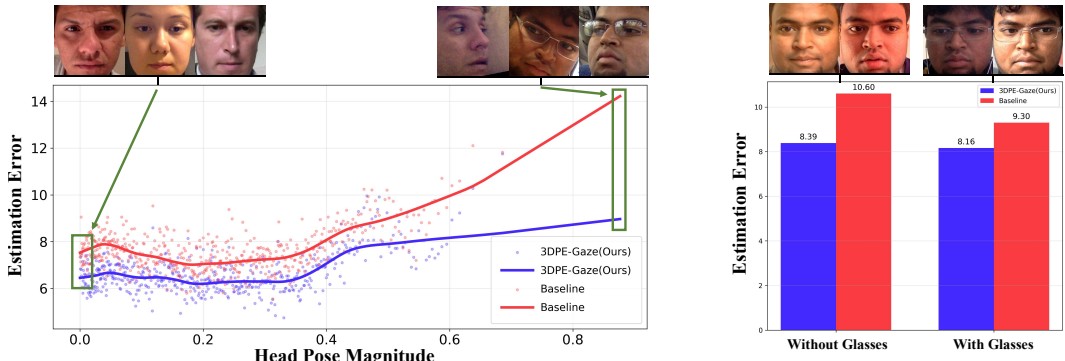

Figure 4: Robustness analysis under extreme scenarios. Left: Gaze estimation accuracy under large head rotations; Right: Impact of glasses occlusion on gaze estimation precision.

**Large Head Rotations.** Head pose variation is a major challenge in gaze estimation. As shown in Figure 4 (left), our method outperforms the baseline across all head poses, with the advantage growing at larger angles. By effectively separating the compound effects of head pose and eyeball movement, our method maintains stable performance even in extreme poses where the baseline fails.

**Generalization Capability in Glasses Occlusion Scenarios.** As shown in Figure 4 (right), our method significantly outperforms the baseline both without glasses (20.9% error reduction, $10.60° \rightarrow 8.39°$) and with glasses (12.3% error reduction, $9.30° \rightarrow 8.16°$). Most importantly, our model demonstrates excellent cross-condition stability, with an error difference of only $0.23°$ between the two conditions, compared to $1.30°$ for the baseline, ensuring a more reliable user experience.

## 5 Conclusion

This paper introduces 3DPE-Gaze, a novel framework for cross-domain gaze estimation that integrates 3D geometric and semantic priors. By leveraging a 3DGP module for geometric decoupling and a SCAM module for semantic purification, our method shifts the task from an unstable appearance-based space to a more robust geometric and semantic one. This design achieves state-of-the-art cross-domain performance without requiring any target domain data.

**Limitations and Future Work.** Limitations of our work include the FLAME model's difficulty in precisely modeling fine eye details and our reliance on a predefined set of semantic concepts for contrastive learning. Future work will focus on three areas: developing specialized parametric eye models for finer detail; exploring adaptive semantic learning strategies, such as concept generation; extending the framework to dynamic, real-time applications on mobile and AR/VR devices.

**Acknowledgement.** This research has been supported by Beijing Natural Science Foundation (L242019).

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
