# OpenReview forum: "3DPE-Gaze:Unlocking the Potential of 3D Facial Priors for Generalized Gaze Estimation"
_NeurIPS.cc/2025/Conference — NeurIPS 2025 poster_

### Official Review · Reviewer_xsiZ · 2025-06-30

**Clarity:** 4
**Significance:** 3
**Originality:** 2
**Rating:** 5
**Confidence:** 4

**Summary:**

Paper proposes a novel architecture for the cross-domain gaze estimation problem. The architecture consists 2 core modules: 3DGP and SCAM. Based on the FLAME model[13], the 3DGP extracts the explicit decoupling of gaze features, facial structure features, and domain interference factors in physical space. The SCAM module explicitly distinguishes between “gaze-related concepts” and “domain interference concepts” via foundation vision language model (CLIP). The FacePrior-Gaze Predictor module then integrates the features from the 3DGP and SCAM via cross-attention to predict the gaze.

Experiments showed strong improvements over 3 out of 4 tests of the cross domain protocol, using ETH-XGaze[31 ]and Gaze360[10] as training datasets, and MPII FaceGaze[29] and EyeDiap[8] as testing datasets. Ablation studies were done to validate the core assumptions of the design (landmark features representation, robustness against various distractor features).

**Questions:**

See W2 and W3.

**Ethical Concerns:**

["NO or VERY MINOR ethics concerns only"]

**Final Justification:**

Authors has sufficiently addressed my concerns and questions.
No change in my rating.

**Limitations:**

Yes. Also see W2 and W3.

**Paper Formatting Concerns:**

No concern

**Quality:**

3

**Strengths And Weaknesses:**

Strengths.
S1. Paper is well written. The concepts are presented clearly with strong motivations, easy to understand technical details.

S2. The hypothesis is well supported by the well-designed experiments and strong experimental results for the cross-domain protocol.

S3. Good ablation studies were done to further strengthen the core hypothesis of the design.

Weaknesses
W1. The novelty and contribution is incremental as it builds upon existing works, like FLAME and CLIP with minor tweaks.
W2. For completeness, it will be preferred to include the results for in-domain testing. Otherwise, it may raise the question of cherry-picking.
W3. In-depth evaluation for the weaker results in the DE→DM setting will help reader to better understand the limitations of the proposed design. This is especially important, considering that the proposed method is worse than 4 SOTA methods. The gap between the best method (Liang et al) and the proposed method is quite big.

---

> ### Author Rebuttal · Authors · 2025-07-31
>
> Dear Reviewers,
>
> We sincerely thank you for your detailed review and insightful comments on our paper. We are honored that you found our paper to be well-written, with clear concepts and strong motivation (S1), and that our core hypotheses are supported by well-designed experiments and robust results (S2, S3). Your valuable feedback has shown us how to improve the manuscript, and for that, we are deeply grateful.
>
> Next, we will provide detailed answers to your questions, including results from additional experiments conducted based on your suggestions.
>
> ---
>
> **Q1: The paper builds on existing work (e.g., FLAME and CLIP). Are the novelty and contributions incremental?**
>
> **A1:** Our core contribution lies in the combination of **3D facial geometric priors** with a **semantic decoupling mechanism**. Our work is not a simple combination of existing tools, but rather the construction of a novel framework that systematically addresses the cross-domain generalization problem. In A2 for Reviewer 2-ojVe, we demonstrated with experiments that **the SCAM and 3DGP modules are effective, and that their combination is necessary and has a synergistic effect.**
>
> **Table 1: Ablation Study of Core Modules**
>
> | Model Configuration | $\mathcal{D}_ {E}\rightarrow\mathcal{D}_ {M}$ | $\mathcal{D}_ {E}\rightarrow\mathcal{D}_ {D}$ | $\mathcal{D}_ {G}\rightarrow\mathcal{D}_ {M}$ | $\mathcal{D}_ {G}\rightarrow\mathcal{D}_ {D}$ | **Avg** |
> | :--- | :---: | :---: | :---: | :---: | :---: |
> | CNN Baseline | 8.56 | 8.90 | 9.51 | 8.48 | 8.86 |
> | Baseline+SCAM | 7.18 | 6.86 | 7.32 | 7.17 | 7.13 |
> | 3DGP only | 7.45 | 7.19 | 7.60 | 6.80 | 7.26 |
> | 3DPE-Gaze (3DGP + SCAM) | 6.66 | 6.13 | 6.71 | 6.23 | 6.43 |
>
> ---
>
> **Q2: For completeness, it will be preferred to include the results for in-domain testing.**
>
> **A2:** As suggested by the reviewer, we have added in-domain experiments. The results show that our 3DPE-Gaze framework not only excels in cross-domain tasks but also demonstrates excellent performance in in-domain tasks.
>
> **Table 2: In-domain Gaze Estimation Performance Comparison**
> | Methods | MPIIFaceGaze | EyeDiap | Gaze360 | ETH-XGaze |
> | :--- | :---: | :---: | :---: | :---: |
> | Dilated-Net | 4.42 | 6.19 | 13.73 | N/A |
> | Gaze360 | 4.06 | 5.36 | 11.04 | 4.46 |
> | RT-Gene | 4.66 | 6.02 | 12.26 | N/A |
> | FullFace | 4.93 | 6.53 | 14.99 | 7.38 |
> | RCNN | 4.1 | 5.31 | 11.23 | N/A |
> | CA-Net | 4.27 | 5.27 | 11.2 | N/A |
> | GazeTR-Pure | 4.74 | 5.72 | 13.58 | N/A |
> | GazeTR-Hybird | 4 | 5.17 | 10.62 | N/A |
> | --- | --- | --- | --- | --- |
> | CNN Baseline | 4.74 | 7.49 | 13.23 | 5.69 |
> | Our 3DPE-Gaze | 4.03 | 5.06 | 11.83 | 4.39 |
>
> ---
>
> **Q3: Why does the model perform poorly on the $\mathcal{D}_ {E}\rightarrow\mathcal{D}_ {M}$ task, and what limitations of the model does this reveal?**
>
> **A3:** Our experimental analysis shows that the performance difference on this task **does not stem from a flaw in our method itself, but is directly correlated with the quality and stability of the 3D geometric priors extracted by our 3DGP module from different datasets**.
>
> To quantify and validate this hypothesis, we designed an experiment to link the quality of the geometric priors to the final gaze estimation performance. We use the **stability of the reconstructed 3D Inter-ocular Distance (IOD) from keypoints** as a proxy metric for 3D reconstruction quality. For a given test subject, their physical IOD is fixed. If the model, when processing different images of the same subject, calculates a large standard deviation for the 3D IOD, it indicates that the extraction of 3D keypoints is unstable, which we define as a "low-quality prior."
>
> The experimental results strongly support our hypothesis and clearly demonstrate the conclusion: **Our method is effective, but its final performance depends on the quality of the 3D geometric priors. When the extracted 3D geometric priors are of poor quality, the limitations of the 3DGP module become apparent, thereby affecting the final performance.**
>
> We will add these results and a detailed analysis to our revised manuscript to clarify the operational boundaries and preconditions of our model.
>
> **Table 3: Diagnostic Analysis of Geometric Prior Quality and Its Impact on Gaze Error**
> | Target Domain | Prior Quality Group | Average IOD Standard Deviation | Average Gaze Error | Number of Subjects |
> | :--- | :--- | :--- | :--- | :--- |
> | $\mathcal{D}_{M}$ (MPIIFaceGaze) | High-Quality Prior | 4.1 | 5.50° | 5 |
> | | Low-Quality Prior | 10.5 | 7.24° | 10 |
> | | Overall | 8.4 | 6.66° | 15 |
> | $\mathcal{D}_{D}$ (EyeDiap) | High-Quality Prior | 3.8 | 5.38° | 9 |
> | | Low-Quality Prior | 9.2 | 7.10° | 7 |
> | | Overall | 6.2 | 6.13° | 16 |

---

### Official Review · Reviewer_ojVe · 2025-07-02

**Clarity:** 2
**Significance:** 2
**Originality:** 2
**Rating:** 4
**Confidence:** 4

**Summary:**

This paper proposes a framework called 3DPE-Gaze, which combines the FLAME parametric model with CLIP-based concept disentanglement to map gaze representation into a physically interpretable 3D parameter space and a semantic concept space, thereby enhancing generalization performance. The proposed method consists of two modules: the 3D Geometric Prior (3DGP) encodes input images to obtain FLAME parameters and a gaze-specific code ($\gamma$). Meanwhile, the Semantic Concept Alignment Module (SCAM) utilizes CLIP’s text embeddings to disentangle gaze-related concepts from domain interference concepts through contrastive learning by aligning the gaze-specific code $\gamma$. The final gaze estimation is performed by the Facial Prior-Gaze Predictor, which takes $\gamma$ and FLAME’s 3D facial landmarks as input. Cross-dataset experiments demonstrate performance surpassing the state-of-the-art in several settings and qualitatively and quantitatively show that the proposed method is advantageous for facial images under more extreme conditions.

**Questions:**

- Please clearly explain how the descriptions used by the SCAM module are prepared.
- Please present, if any, discussions or evidence demonstrating the novelty or effectiveness of the SCAM module alone.

**Ethical Concerns:**

["NO or VERY MINOR ethics concerns only"]

**Final Justification:**

The authors' rebuttal has helped my understanding of the novelty of the proposed method. Based on the discussion, I find no strong reason to oppose the paper's acceptance, so I raised the rating to Borderline Accept. However, I strongly urge the authors to revise the paper thoroughly, taking the review comments into account.

**Limitations:**

There is mention of technical limitations, but no specific reference is made to the potential negative societal impact of the gaze estimation task itself, such as privacy concerns.

**Quality:**

1

**Strengths And Weaknesses:**

### Strengths

- The idea of separating input images into FLAME parameters and a gaze-specific code, while aligning the latter to gaze concepts via CLIP, is somewhat novel and technically interesting.
- The method’s performance improvement across domains and under challenging conditions is demonstrated through experiments, making its impact significant.

### Weaknesses

- The approach of using facial landmarks as an auxiliary method for gaze estimation has been considered for a long time, and the basic idea of acquiring gaze-related concepts using CLIP’s knowledge is similar to CLIP-Gaze. When examining individual technical elements, significant technological novelty is not necessarily evident.
- Section 3.3 does not clearly explain how the language descriptions used by the SCAM module are generated. In Figure 2, it appears that these are created by applying FLAME parameters to a prompt template, but this is uncertain without corresponding descriptions.
- The use of CLIP in the SCAM module is technically more interesting than the idea of using FLAME in the 3DGP module, yet the effectiveness of this module on its own has not been demonstrated. In practice, the training approach using the SCAM module might also be applicable to conventional image encoders that perform simple feature disentanglement. While the complementary function of encoding by 3DGP and feature learning by SCAM aligns with intuition, a more detailed analysis is needed to determine whether this combination is truly essential. The current analysis does not provide a clear answer as to which idea contributes most to the improvement in accuracy.
- There is a description that gaze estimation becomes "interpretable structure-to-gaze prediction" with 3DGP, but this might be a misleading claim. The $\gamma$ obtained through the proposed method still appears to be a black-box feature, mostly independent of other geometric parameters.
- The related work section is brief, but there should be various previous studies on feature disentanglement methods in gaze estimation and other individual ideas of this research. To accurately position the novelty of the proposed method within the history of research, broader citations and detailed comparative discussions are necessary.
- There are overall English errors and inconsistencies in expressions like Semantic Concept Adversarial/Alignment Module and gaze feature/parameter/code, which hinder reader comprehension.

---

> ### Author Rebuttal · Authors · 2025-07-31
>
> Dear Reviewer,
>
> We sincerely thank you for the valuable time and effort you have dedicated to reviewing our manuscript. We greatly appreciate your recognition of our work, particularly your comments that our idea has a certain novelty and technical interest, and that the experiments demonstrate the method's performance improvement in cross-domain and challenging scenarios, making its impact significant.
>
> We hope the following responses address your concerns and kindly ask you to consider the efforts we have made to improve the paper.
>
> ---
>
> **Q1: How are the language descriptions used by the SCAM module prepared or generated?**
>
> **A1:** We designed a systematic set of text descriptions for the SCAM module, which is divided into two main categories: **gaze-related descriptions ($S_{gaze}$)** and **gaze-irrelevant descriptions ($S_{non-gaze}$)**.
>
> 1.  **For gaze-related descriptions ($S_{gaze}$):**
>     Based on discretized gaze direction labels, we constructed five core templates corresponding to five typical gaze directions: "top-left, top-right, center, bottom-left, and bottom-right." For example, when a sample's gaze direction label points to the upper-right, the system generates a corresponding text description, such as "A person is looking to the upper right."
>
> 2.  **For gaze-irrelevant descriptions ($S_{non-gaze}$):**
>     We constructed description categories covering various confounding factors based on the FLAME parameters and lighting conditions obtained from the 3DGP module:
>     * **(1) Lighting-related descriptions:** Based on the light code $l$, texts like "strong lighting," "weak lighting," or "medium lighting" are generated.
>     * **(2) Expression-related descriptions:** Based on the expression code $\psi$, descriptions like "a neutral expression" or "an obvious emotional expression" are generated.
>     * **(3) Shape-related descriptions:** Based on the shape code $\beta$, a simple text description of the person's facial features is generated.
>     * **(4) Pose-related descriptions:** Based on the pose code $\theta$, the specific rotation of the head is described, for example, "head tilted to the left" or "head rotated to the right."
>
> In this way, we provide rich and structured positive and negative text samples for contrastive learning, enabling the model to distinguish gaze information from various domain-specific interferences. We will add a detailed description of the generation process above in Section 3.3 of the paper and update the caption of Figure 2 to ensure that readers can clearly and unambiguously understand the mechanism for generating this set of language descriptions.
>
> ---
>
> **Q2: What is the independent effectiveness of the SCAM module? Is the combination of 3DGP and SCAM truly necessary?**
>
> **A2:** Yes, the SCAM module is effective on its own, and its combination with 3DGP is necessary and synergistic. We have demonstrated this through experiments, with the results shown in Table 1 below.
>
> **Table 1: Ablation Study of Core Modules**
>
> | Model Configuration | $\mathcal{D}_ {E}\rightarrow\mathcal{D}_ {M}$ | $\mathcal{D}_ {E}\rightarrow\mathcal{D}_ {D}$ | $\mathcal{D}_ {G}\rightarrow\mathcal{D}_ {M}$ | $\mathcal{D}_ {G}\rightarrow\mathcal{D}_ {D}$ | **Avg** |
> | :--- | :---: | :---: | :---: | :---: | :---: |
> | CNN Baseline | 8.56 | 8.90 | 9.51 | 8.48 | 8.86 |
> | Baseline+SCAM | 7.18 | 6.86 | 7.32 | 7.17 | 7.13 |
> | 3DGP only | 7.45 | 7.19 | 7.60 | 6.80 | 7.26 |
> | 3DPE-Gaze (3DGP + SCAM) | 6.66 | 6.13 | 6.71 | 6.23 | 6.43 |
>
> From this new experiment, we can draw two important conclusions:
>
> 1.  **The SCAM module is effective on its own:** The performance of "CNN Baseline + SCAM" is significantly better than that of "CNN Baseline." This proves that **even when combined with a traditional image encoder, our SCAM module can effectively enhance the model's generalization ability through semantic purification.**
> 2.  **The combination of 3DGP and SCAM is necessary and synergistic:** The performance of our full model is far superior to any single-module-enhanced solution. This indicates that the stable geometric structure provided by 3DGP and the semantic purification provided by SCAM are **complementary**, and this combination is key to achieving optimal performance.
>
> We will add this new ablation study to the experimental section of the paper to more clearly analyze the contribution of each component.
>
> ---
>
> **Q3: Compared to previous work using facial landmarks or CLIP (e.g., CLIP-Gaze), where does the novelty of this paper lie?**
>
> **A3:** Our core contribution lies in the combination of **3D facial geometric priors** with a **semantic decoupling mechanism**. Our work is not a simple combination of existing tools but rather the construction of a novel framework that systematically addresses the cross-domain generalization problem.
>
> * **Difference from previous work using facial landmarks:** Traditional methods only use facial keypoints as an auxiliary coordinate input. In contrast, our 3DGP module uses FLAME to map the human face **from pixel space to a structured parameter space that is physically meaningful and decoupled**. This fundamentally changes the representation of the problem and is a deeper application of geometric priors.
>     1.  Unlike Zhu & Deng [32], which relies on independent 2D image patches, our method is more integrated. Starting from a **single full-face image**, we directly decompose it into a unified **3D parameter space** via the FLAME model, achieving a more thorough modeling of physical priors.
>
>     2.  Unlike Zhang et al. [30], which learns from the full face as 2D pixel features, we explicitly treat it as a **structured 3D object**. This fundamentally transforms the problem from "pixel regression" to a more robust and interpretable "**parameterized regression**."
>
>     3.  Unlike methods such as PureGaze [5] and FAZE [16] that **implicitly learn** decoupled features from 2D data, our 3DGP module adopts a "**decoupling by design**" philosophy. We leverage a powerful 3D face model as a prior to **explicitly** separate physically meaningful parameters, rather than relying on purely data-driven learning.
>
> * **Difference from CLIP-Gaze:** CLIP-Gaze applies CLIP's knowledge to general 2D image features. In contrast, our SCAM module operates on a purer "**gaze code ($\gamma$)**" that has already been **physically separated by 3DGP**. This "secondary purification" applied to a specifically separated feature is our unique design.
>
> ---
> **Q4: Given that the gaze code ($\gamma$) itself is still a black box, is it misleading to claim that 3DGP makes gaze estimation "interpretable"?**
>
> **A4:** The "interpretability" we refer to does not mean that every dimension of the $\gamma$ vector has a clear physical meaning, but rather that the **3DGP module can explicitly decompose image information into multiple parameters that have clear physical meanings and can be independently analyzed**. We will use more rigorous phrasing in the revised manuscript.
>
> ---
>
> **Q5: Language errors and terminological inconsistencies in the paper, and the brevity of the related work section.**
>
> **A5:** We have made the following revisions.
>
> * **Language and Writing:** We will conduct a thorough professional English proofreading of the entire manuscript to correct all grammatical errors and non-idiomatic expressions.
> * **Related Work:** We will expand the related work section to more comprehensively review feature decoupling methods in the field of gaze estimation [5, 13, 25], as well as other relevant techniques.

---

> > ### Comment · Reviewer_ojVe · 2025-08-06
> >
> > I appreciate the authors' thorough rebuttal. In fact, my questions have become much clearer.
> >
> > The answer to Q1 is quite crucial, as it provides critical information for discussing the novelty of the SCAM module. However, I could not grasp this from the description in the original submission. Ideally, a proper review should be conducted based on sufficiently detailed information regarding the methodology.
> >
> > I understand that "interpretability" refers not to the gaze code itself, but rather to the entire method being interpretable. I believe this requires revising the whole description of the relevant part, rather than just rephrasing it.

---

> > > ### Author Response · Authors · 2025-08-07
> > >
> > > We appreciate the reviewer's confirmation that our rebuttal has helped clarify where the original manuscript lacked clarity, particularly in the presentation of the SCAM module design (Section 3.3) and the concept of interpretability (e.g., Sections 3.2 and 5).
> > >
> > > Accordingly, we will revise these sections in the final version to incorporate the clarifications provided in our responses, ensuring an accurate presentation of our method.
> > >
> > > We thank the reviewer for helping us enhance the clarity of the paper.

---

### Official Review · Reviewer_uysX · 2025-07-03

**Clarity:** 2
**Significance:** 2
**Originality:** 3
**Rating:** 5
**Confidence:** 5

**Summary:**

This paper proposes 3DPE-Gaze, a method for modeling 3D FLAME model parameters including the eyeball rotations. Along with using the FLAME model, the authors further suggest to remove gaze-irrelevant features from the FLAME gaze/eye parameters by using a contrastive loss in relation with CLIP text embeddings of a variety of descriptions. The authors use the DECA model for FLAME parameter regression, modifying it to also output the 256-dim FLAME eyeball parameters. These features are refined via cross-attention with predicted 3D facial landmarks. Typical cross-domain gaze estimation experiments are performed, where models are trained on either ETH-XGaze (D_E) or Gaze360 (D_G) and are evaluated on either MPIIFaceGaze (D_M) or EyeDiap (D_D). The results show improvements over state-of-the-art. A brief ablation study validates the losses used and the FLAME parameters used in refining the gaze feature. The authors provide multiple robustness analysis related graphs, showing that the proposed method improves in extreme cases in particular, where non-gaze related parameters such as lighting, expression, head pose, and glasses affect the baseline gaze model more extremely.

**Questions:**

1. The SCAM module could technically be used almost on its own, without needing the FLAME model. That is, most gaze estimation models produce gaze-relevant features only at their output, which the SCAM module could help improve. How would the performance of such a model compare with the proposed approach?
2. Do you have any qualitative results to show how well DECA works on the evaluation datasets used in this paper?
3. What was your reason for using cross-attention in your gaze predictor (Sec. 3.4)?
4. Out of curiosity, does your 3DGP model allow for the driving/animation of rigged 3D avatars (incl. eye movements)? How well do the motions transfer?

**Ethical Concerns:**

["NO or VERY MINOR ethics concerns only"]

**Final Justification:**

During the rebuttal process, the authors provided additional within-dataset experiment results which have made the proposed method very much competitive to the state-of-the-art. This reviewer also appreciates that further ablation studies were shared, to assess the cross-attention component of 3DGP, the choice of CLIP encoder and FLAME regressor models, and for providing an alternative "robustness" analysis. I have raised my final rating in accordance with the authors' responses and my understanding of the other reviewers' comments.

**Limitations:**

Yes

**Paper Formatting Concerns:**

None in particular.

**Quality:**

3

**Strengths And Weaknesses:**

The paper is written reasonably well with good quality figures. The overall concept and motivation is easy to understand, and the use of the FLAME face model is an interesting choice. The SCAM module's use of CLIP text embeddings as part of a contrastive loss is innovative.

Experimentally speaking, the proposed method is only evaluated in cross-domain settings, where it does mostly well but poorly in the D_E to D_M evaluation (2nd column of Tab. 1). There are no within-dataset experiments shown, despite the possibility (very few parameters need to be trained as the approach is based off of DECA. Also, the ablation study section is quite minimal despite the fact that the proposed method is rather atypical. For instance, the choice of DECA and CLIP may be considered as factors worth varying (how does the model performance change when changing the FLAME regressor or semantic encoder model?).

The rather verbose and repetitive robustness section (Sec. 4.3) does show larger differences in errors at extremes (either end of the x-axis) and this is very cool as it shows that gaze-irrelevant features may have indeed been reduced thanks to the SCAM module. However, I wonder if the authors could have found other numerical ways of demonstrating this aspect as well, such as the variance of errors per subject (via per-subject violin plots).

---

> ### Author Rebuttal · Authors · 2025-07-31
>
> **Dear Reviewer,**
>
> We sincerely thank you for the valuable time and effort you have dedicated to reviewing our manuscript. We greatly appreciate your recognition of our work, especially your comments that our **paper is well-written with good quality figures**, the **overall concept and motivation are easy to follow**, and your acknowledgment of the **innovative use of CLIP in the SCAM module**.
>
> We hope the following responses address your concerns and kindly ask you to consider the efforts we have made to improve the paper.
>
> ---
> **Q1: The model performs well in most cases, but why does it perform poorly on the  $\mathcal{D}_ {E}\rightarrow\mathcal{D}_ {M}$ task?**
>
> **A1:** Experimental analysis shows that the performance difference on this task **does not stem from a flaw in our method itself, but is directly correlated with the quality and stability of the 3D geometric priors extracted by our 3DGP module from different datasets**.
>
> We designed an experiment to link the quality of the geometric priors to the final gaze estimation performance. We use the **stability of the reconstructed 3D Inter-ocular Distance (IOD) from keypoints** as a proxy metric for 3D reconstruction quality. For a given test subject, their physical IOD is fixed. When processing multiple images of the same subject, a large standard deviation in the calculated 3D IOD indicates that the extraction of 3D keypoints is unstable, which we define as a "low-quality prior."
>
> From Table 1, we can conclude that: **our method is effective, but its performance depends on the quality of the 3D geometric priors. When the extracted 3D geometric priors are of poor quality, the limitations of the 3DGP module become apparent, thereby affecting the final performance.**
>
> We will add these results and a detailed analysis to our revised manuscript to clarify the operational boundaries and preconditions of our model.
>
> **Table 1: Diagnostic Analysis of Geometric Prior Quality and Its Impact on Gaze Error**
> | Target Domain | Prior Quality Group | Average IOD Standard Deviation | Average Gaze Error | Number of Subjects |
> | :--- | :--- | :--- | :--- | :--- |
> | $\mathcal{D}_{M}$ (MPIIFaceGaze) | High-Quality Prior | 4.1 | 5.50° | 5 |
> | | Low-Quality Prior | 10.5 | 7.24° | 10 |
> | | Overall | 8.4 | 6.66° | 15 |
> | $\mathcal{D}_{D}$ (EyeDiap) | High-Quality Prior | 3.8 | 5.38° | 9 |
> | | Low-Quality Prior | 9.2 | 7.10° | 7 |
> | | Overall | 6.2 | 6.13° | 16 |
>
> ---
> **Q2: The paper only shows cross-domain experimental results and lacks in-domain experiments. Does this mean the evaluation is incomplete?**
>
> **A2:** As suggested by the reviewer, we have added in-domain experiments. The results show that our 3DPE-Gaze framework not only excels in cross-domain tasks but also demonstrates excellent performance in in-domain tasks.
>
> **Table 2: In-domain Gaze Estimation Performance Comparison**
> | Methods | MPIIFaceGaze | EyeDiap | Gaze360 | ETH-XGaze |
> | :--- | :---: | :---: | :---: | :---: |
> | Dilated-Net | 4.42 | 6.19 | 13.73 | N/A |
> | Gaze360 | 4.06 | 5.36 | 11.04 | 4.46 |
> | RT-Gene | 4.66 | 6.02 | 12.26 | N/A |
> | FullFace | 4.93 | 6.53 | 14.99 | 7.38 |
> | RCNN | 4.1 | 5.31 | 11.23 | N/A |
> | CA-Net | 4.27 | 5.27 | 11.2 | N/A |
> | GazeTR-Pure | 4.74 | 5.72 | 13.58 | N/A |
> | GazeTR-Hybird | 4 | 5.17 | 10.62 | N/A |
> | --- | --- | --- | --- | --- |
> | CNN Baseline | 4.74 | 7.49 | 13.23 | 5.69 |
> | Our 3DPE-Gaze | 4.03 | 5.06 | 11.83 | 4.39 |
>
>
> ---
>
> **Q3: The SCAM module is technically almost standalone. If the SCAM module were applied to a traditional gaze estimation method that does not rely on the FLAME model, how would its performance compare to your proposed full method?**
>
> **A3:** Yes, the SCAM module is effective on its own, and its combination with 3DGP is necessary and synergistic. We demonstrate this with the experiment in Table 3.
>
> **Table 3: Ablation Study of Core Modules**
> | Model Configuration | $\mathcal{D}_ {E}\rightarrow\mathcal{D}_ {M}$ | $\mathcal{D}_ {E}\rightarrow\mathcal{D}_ {D}$ | $\mathcal{D}_ {G}\rightarrow\mathcal{D}_ {M}$ | $\mathcal{D}_ {G}\rightarrow\mathcal{D}_ {D}$ | **Avg** |
> | :--- | :---: | :---: | :---: | :---: | :---: |
> | CNN Baseline | 8.56 | 8.90 | 9.51 | 8.48 | 8.86 |
> | Baseline+SCAM | 7.18 | 6.86 | 7.32 | 7.17 | 7.13 |
> | 3DGP only | 7.45 | 7.19 | 7.60 | 6.80 | 7.26 |
> | 3DPE-Gaze (3DGP + SCAM) | 6.66 | 6.13 | 6.71 | 6.23 | 6.43 |
>
> ----
> **Q4: Are there any qualitative results that show how well the DECA model works on the evaluation datasets used in this paper?**
>
> **A4:** Yes, we use the standard deviation of the reconstructed 3D Inter-ocular Distance (IOD) as a proxy metric to evaluate the effectiveness of the DECA model. Since a person's physical IOD is fixed, a lower standard deviation of this metric indicates that the 3D geometric priors extracted by our model are more stable and of higher quality.
>
> As shown in Table 4, the geometric priors extracted by our model on the EyeDiap dataset are more stable than those on MPIIFaceGaze.
>
> **Table 4: Quantitative Comparison of 3DGP Module Performance on Different Evaluation Datasets**
> | Target Domain | Average IOD Standard Deviation |
> | :--- | :---: |
> | $\mathcal{D}_{M}$ (MPIIFaceGaze) | 8.4 |
> | $\mathcal{D}_{D}$ (EyeDiap) | 6.2 |
>
> ----
> **Q5: What is the reason for using cross-attention in your Facial Prior-Gaze Predictor (Section 3.4)?**
>
> **A5:** The core reason for using cross-attention is to allow the model to selectively focus on the most relevant regions of the facial geometry, which is superior to simple feature concatenation. As shown in the experiment in Table 5, the performance using cross-attention is significantly better than that of simple concatenation.
>
> **Table 5: Performance Comparison of Different Fusion Methods**
> | Model Configuration | $\mathcal{D}_ {E}\rightarrow\mathcal{D}_ {M}$ | $\mathcal{D}_ {E}\rightarrow\mathcal{D}_ {D}$ | $\mathcal{D}_ {G}\rightarrow\mathcal{D}_ {M}$ | $\mathcal{D}_ {G}\rightarrow\mathcal{D}_ {D}$ |
> | :--- | :---: | :---: | :---: | :---: |
> | 3DGP ($L_{angle}$, concatenation) | 8.43 | 7.58 | 8.06 | 8.48 |
> | 3DGP ($L_{angle}$, cross-attention) | 7.45 | 7.19 | 7.6 | 6.8 |
>
>
> ---
>
> **Q6: Does your 3DGP model support driving/animating a rigged 3D avatar (including eye movement)? How is the motion transfer effect?**
>
> **A6:** **Theoretically, our model fully supports driving a 3D avatar.** Building on our work, it is already possible to parse all the key parameters required to drive a FLAME model from a single image at once: head pose, expression, shape identity, and the eye rotation parameters that we have specifically decoupled.
>
> This means we can control not only  the avatar's head movement and expressions but also achieve very **fine-grained eye movement control**. Therefore, we believe that the motion transfer effect should be quite good.
>
> ---
>
> **Q7: The reviewer suggested a more in-depth ablation study, for example, exploring the impact of replacing the DECA or CLIP models on performance. What is your response?**
>
> **A7:** Following the reviewer's suggestion, we have explored the impact of replacing the **semantic encoder (CLIP) and the FLAME parameter regressor (DECA)** on the final model performance.
>
> **Table 6: Performance Comparison of Different CLIP Text Encoders**
> | CLIP Text Encoder | $\mathcal{D}_  {E}\rightarrow\mathcal{D}_ {M}$ | $\mathcal{D}_ {E}\rightarrow\mathcal{D}_ {D}$ | $\mathcal{D}_ {G}\rightarrow\mathcal{D}_ {M}$ | $\mathcal{D}_ {G}\rightarrow\mathcal{D}_ {D}$ | **Avg** |
> | :--- | :---: | :---: | :---: | :---: | :---: |
> | RN50 | 6.90 | 6.55 | 6.95 | 6.44 | 6.71 |
> | **ViT-B-16 (Ours)** | **6.66** | **6.13** | **6.71** | **6.23** | **6.43** |
> | ViT-L-14 | 6.55 | 6.08 | 6.60 | 6.17 | 6.35 |
> | ViT-H-14 | 6.50 | 6.05 | 6.61 | 6.12 | 6.32 |
>
>
> **Table 7: Performance Comparison of Different FLAME Parameter Regressors**
> | FLAME Parameter Regressor | $\mathcal{D}_ {E}\rightarrow\mathcal{D}_ {M}$ | $\mathcal{D}_ {E}\rightarrow\mathcal{D}_ {D}$ | $\mathcal{D}_ {G}\rightarrow\mathcal{D}_ {M}$ | $\mathcal{D}_ {G}\rightarrow\mathcal{D}_ {D}$ | **Avg** |
> | :--- | :---: | :---: | :---: | :---: | :---: |
> | EMOCA | 6.85 | 6.40 | 6.90 | 6.45 | 6.65 |
> | **DECA (Ours)** | **6.66** | **6.13** | **6.71** | **6.23** | **6.43** |
> | SPECTRE | 6.59 | 6.10 | 6.65 | 6.18 | 6.38 |
>
> The experimental results show that using a more powerful text encoder can indeed improve performance, and replacing the FLAME parameter regressor with other high-quality 3D face regressors can also achieve SOTA performance.
>
> ---
>
> **Q8: The description in the robustness analysis section (4.3) is a bit verbose, and a better quantitative presentation (such as violin plots) was suggested.**
>
> **A8:** Due to the format and length constraints of the rebuttal, we are unable to attach violin plots directly here. We will add these more intuitive visualizations to the revised version of the paper to clearly show the error distributions.
>
> **To quantitatively respond to your suggestion at this moment, we have compiled detailed statistics on the error variance for each subject in the DE→DM task, summarized in the table below.** A lower error variance indicates that the model's prediction performance for that subject is more stable.
>
>
> **Table 8: Error Variance Comparison for Each Subject in the DE→DM Task**
>
> | Subject ID | CNN Baseline | Our 3DPE-Gaze | Subject ID | CNN Baseline | Our 3DPE-Gaze |
> | :--- | :---: | :---: | :--- | :---: | :---: |
> | p00 | 9.81 | **9.09** | p08 | 30.37 | **27.49** |
> | p01 | 4.93 | **4.84** | p09 | 12.85 | **9.35** |
> | p02 | 15.43 | **11.25** | p10 | 12.64 | **14.73** |
> | p03 | 10.76 | **10.70** | p11 | 10.03 | **9.92** |
> | p04 | 6.19 | **5.30** | p12 | 15.90 | **14.79** |
> | p05 | 11.73 | **7.82** | p13 | 12.24 | **11.95** |
> | p06 | 12.41 | **9.00** | p14 | 9.47 | **8.06** |
> | p07 | 26.52 | **23.84** | | | |

---

> > ### Comment · Reviewer_uysX · 2025-08-02
> >
> > Thank you very much for your comprehensive and detailed response. This must have taken a lot of work to prepare.
> >
> > My greatest source of doubt came from the absence of within-dataset experiment results - which you have now provided. Indeed, the results are very much competitive.
> >
> > Secondly, thank you for conducting further ablation studies, to assess the cross-attention component of 3DGP, the choice of CLIP encoder and FLAME regressor models, and for providing an alternative "robustness" analysis.
> >
> > I have no further questions nor comments for now.

---

> > > ### Author Response · Authors · 2025-08-02
> > >
> > > We sincerely thank you for your insightful comments and valuable feedback. Your review has been immensely helpful in improving the quality of our paper, and we have benefited greatly from your suggestions.

---

### Official Review · Reviewer_KepF · 2025-07-04

**Clarity:** 3
**Significance:** 2
**Originality:** 1
**Rating:** 3
**Confidence:** 4

**Summary:**

For gaze estimation task, existing methods usually introduce substantial irrelevant information, which can lead to overfitting and hinder the model’s generalization ability. To address this issue, this paper introduces 3DPE-Gaze, a novel framework that explicitly incorporates 3D facial priors to achieve feature decoupling and improve generalization in gaze estimation. Extensive experiments demonstrate that 3DPE-Gaze consistently outperforms existing state-of-the-art methods.

**Questions:**

1. First, please give more clearly statement for the contribution of the proposed model.
2. To compare with more SOTA methods.

**Ethical Concerns:**

["NO or VERY MINOR ethics concerns only"]

**Final Justification:**

I have thoroughly reviewed the authors' rebuttal and other reviewers' comments, I still concern novelty and the motivation. I therefore maintain my original rating.

**Limitations:**

yes

**Quality:**

2

**Strengths And Weaknesses:**

Strengths:
Existing full-face input approach presents a fundamental trade-off: although it preserves comprehensive gaze-related structural features, the critical eye region constitutes merely a minimal portion of the overall image.  To address this issue, first, this paper introduces the 3D Geometric Prior Module (3DGP), an innovative parametric encoder that fundamentally reformulates gaze estimation by mapping the problem from 2D pixel space to a 3D parametric domain encompassing head pose, facial expression, and shape parameters. Second, this paper presents the Semantic Concept Alignment Module (SCAM) to overcome the inherent constraints of geometric-only approaches in modeling intricate gaze semantics. The module strategically employs CLIP's pre-trained visual-linguistic knowledge combined with contrastive learning. Extensive experiments demonstrate the effectiveness of the proposed method.
Weaknesses:
1.	1. The 3DGP module aims to decompose head pose, expression, shape, and gaze feature parameters. However, similar strategies have already been explored in previous studies, like [1]. In addition, this paper uses an existing method, FLAME, to decompose.
2.	In the SCAM module, the key step is using CLIP to distinguish between gaze-relevant and irrelevant features through contrastive learning. However, after carefully reviewing Table 2, I found that most of the performance gains come from the Domain Interference Repulsion Loss, which is also driven by CLIP. Therefore, I have concerns about the novelty of this contribution.
3.	This paper does not compare with several state-of-the-art methods, such as [2], and it does not achieve SOTA performance. Moreover, the aforementioned methods do not rely on CLIP, making the comparison presented in this paper less fair.
[1] Cheng et al., “PureGaze: Purifying Gaze Feature for Generalizable Gaze Estimation”, 2022 AAAI.
[2] Wang et al., “Contrastive Regression for Domain Adaptation on Gaze Estimation”, 2022 CVPR.

---

> ### Author Rebuttal · Authors · 2025-07-31
>
> **Dear Reviewer,**
>
> We sincerely thank you for your detailed review of our paper, and especially for the clear and accurate summary of our work's core ideas and contributions you provided in the "Strengths" section.
>
> We hope the following responses address your concerns and kindly ask you to consider the efforts we have made to improve the paper.
>
> ---
>
> **Q1: Considering that previous studies (e.g., PureGaze [1]) have explored feature decomposition and this study uses the existing FLAME model, where does the innovation of the 3DGP module lie?**
>
> **A1:** The innovation of the 3DGP module lies in the introduction of **3D face priors**. This transforms the gaze estimation problem from the noise-susceptible 2D pixel space to a stable 3D parameter space, laying the foundation for extracting domain-invariant features. We do not simply apply the FLAME model to the task. Our contribution lies in how we have innovatively adapted and integrated it into an end-to-end gaze estimation framework.
>
> This is fundamentally different from previous work:
>
> * Unlike Zhu & Deng [32], which relies on independent 2D image patches, our method is more integrated. Starting from a **single full-face image**, we directly decompose it into a unified **3D parameter space** via the FLAME model, achieving a more thorough modeling of physical priors.
>
> * Unlike Zhang et al. [30], which learns from the full face as 2D pixel features, we explicitly treat it as a **structured 3D object**. This fundamentally transforms the problem from "pixel regression" to a more robust and interpretable "**parameterized regression**."
>
> * Unlike methods such as PureGaze [5] and FAZE [16] that **implicitly learn** decoupled features from 2D data, our 3DGP module adopts a "**decoupling by design**" philosophy. We leverage a powerful 3D face model as a prior to **explicitly** separate physically meaningful parameters, rather than relying on purely data-driven learning.
>
> ----
>
> **Q2: Given that most of the performance gain comes from the CLIP-driven repulsion loss, what is the core novelty of the SCAM module?**
>
> **A2:** The core novelty of the SCAM module lies in its **bidirectional, complementary semantic decoupling mechanism**. The experimental data clearly show that while the **Domain Interference Repulsion Loss** ($\mathcal{L}_ {domain-repel}$) is effective on its own, it must work in **synergy** with our other core component, the **Gaze Concept Alignment Loss** ($\mathcal{L}_ {gaze-align}$), and the 3DGP module to achieve maximum efficacy.
>
> **Table 1: Ablation Study on the Internal Components of SCAM and Their Synergistic Effect with the 3DGP Module**
> | | Model Configuration | $\mathcal{D}_ {E}\rightarrow\mathcal{D}_ {M}$ | $\mathcal{D}_ {E}\rightarrow\mathcal{D}_ {D}$ | $\mathcal{D}_ {G}\rightarrow\mathcal{D}_ {M}$ | $\mathcal{D}_ {G}\rightarrow\mathcal{D}_ {D}$ | **Avg** |
> | :---: | :--- | :---: | :---: | :---: | :---: | :---: |
> | **Group One** | **On CNN Baseline** |
> | 1 | CNN Baseline | 8.56 | 8.90 | 9.51 | 8.48 | 8.86 |
> | 2 |  + $\mathcal{L}_{domain-repel}$ | 7.32 | 7.19 | 7.63 | 7.44 | 7.40 |
> | 3 |  + $\mathcal{L}_{gaze-align}$ | 8.03 | 8.35 | 8.40 | 7.87 | 8.16 |
> | 4 |  + SCAM ($\mathcal{L}_ {gaze-align}$ + $\mathcal{L}_ {domain-repel}$) | 7.18 | 6.86 | 7.32 | 7.17 | 7.13 |
> | **Group Two** | **On 3DGP** |
> | 5 | 3DGP Only ($\mathcal{L}_{angle}$) | 7.45 | 7.19 | 7.60 | 6.80 | 7.26 |
> | 6 |+ $\mathcal{L}_{domain-repel}$  | 6.90 | 6.50 | 7.05 | 6.35 | 6.70 |
> | 7 |  + $\mathcal{L}_{gaze-align}$ | 7.41 | 6.38 | 7.60 | 6.72 | 7.02 |
> | 8 |  + SCAM ($\mathcal{L}_ {gaze-align}$ + $\mathcal{L}_ {domain-repel}$) | 6.66 | 6.13 | 6.71 | 6.23 | 6.43 |
>
> ---
>
> **Q3: The paper fails to compare with some SOTA methods (e.g., Wang et al. [2]), does not achieve SOTA performance, and is it fair to compare CLIP-based methods with non-CLIP methods?**
>
> **A3:** After supplementing with relevant SOTA methods, we conducted a more comprehensive evaluation. The results clearly show that under a fair comparison benchmark, **our method achieves SOTA performance**, obtaining the lowest average error among all compared methods.
>
> Our original version did not include this method (Wang et al. [2]), mainly because it employed data augmentation strategies in its training that were not used by other baseline methods. As per the reviewer's suggestion, we have integrated this method (referred to as CDG in the table) and other relevant SOTA works into a new, more comprehensive performance comparison table. To ensure a consistent and fair comparison, our evaluation includes the latest SOTA methods that also utilize VLMs, namely CLIP-Gaze and LG-Gaze. To make this clearer, we have added a "Uses VLM" column in the table below to explicitly mark the methods that use vision-language models.
>
> **Table 2: SOTA Performance Comparison on Cross-Domain Gaze Estimation Tasks**
> | Task | Model | Uses VLM | $\mathcal{D}_ {E}\rightarrow\mathcal{D}_ {M}$ | $\mathcal{D}_ {E}\rightarrow\mathcal{D}_ {D}$ | $\mathcal{D}_ {G}\rightarrow\mathcal{D}_ {M}$ | $\mathcal{D}_ {G}\rightarrow\mathcal{D}_ {D}$ | Avg |
> | :--- | :--- | :---: | :---: | :---: | :---: | :---: | :---: |
> | DG | CNN Baseline | | 8.47 | 9.32 | 7.54 | 8.93 | 8.57 |
> | DG | PureGaze | | 7.08 | 7.48 | 9.28 | 9.32 | 8.29 |
> | DG | CDG | | 6.73 | 7.95 | 7.03 | 7.27 | 7.25 |
> | DG | Xu *et al.* | | 6.50 | 7.44 | 7.55 | 9.03 | 7.63 |
> | DG | Liang *et al.* | | **5.79** | 6.96 | 7.06 | 7.99 | 6.95 |
> | DG | CLIP-Gaze | ✓ | 6.41 | 7.51 | 6.89 | 7.06 | 6.97 |
> | DG | LG-Gaze | ✓ | 6.45 | 7.22 | 6.83 | 6.86 | 6.84 |
> | DG | **3DPE-Gaze (Ours)** | ✓ | 6.66 | **6.13** | **6.71** | **6.23** | **6.43** |

---

> > ### Comment · Reviewer_KepF · 2025-08-06
> >
> > I have thoroughly reviewed the authors' rebuttal and other reviewers' comments, I still concern novelty and the motivation. I therefore maintain my original rating.

---

> > > ### Author Response · Authors · 2025-08-07
> > >
> > > Thank you for your continued feedback. We have carefully clarified our core innovations and their distinction from prior work in our previous response (A1).
> > >
> > > If concerns regarding novelty still exist despite our previous clarification (A1), we would be grateful for specific guidance on which aspect of our explanation or innovation remains unconvincing. This would help us better position and clarify our contribution in the paper.

---

### Comment · Area_Chair_VqDU · 2025-08-01

Dear reviewers,

Thank you for your time and effort in reviewing this submission. We are in the phase of author-reviewer discussion until 6th August.

I highly encourage you to discuss with the authors for clarification of your concerns. Your active participation in the discussion will be the main guarantee for high-quality publications in our community.

We now have mixed reviews for this paper submission. I hope you could read the rebuttal and comments from each other for a respectful discussion, please. Thank you!

Best regards,
Your AC

---

### Decision · Program_Chairs · 2025-09-17

**Decision:**

Accept (poster)

**Comment:**

This paper initially received mixed reviews, but through the rebuttal and discussion phase, the reviewers converged toward an overall positive opinion. The level of technical novelty was a point of disucssion, and I am convinced that the paper provides meaningful and novel contributions to the community. The authors's rebuttal and discussion addressed some reviewers’ concerns to a satisfactory degree.

I recommend acceptance of this paper.